# World Models as Reference Trajectories for Rapid Motor Adaptation

**Carlos Stein Brito**
NightCity Labs, Champalimaud Centre for the Unknown
Lisbon, Portugal
carlos.stein@nightcitylabs.ai

**Daniel C. McNamee**
Champalimaud Centre for the Unknown
Lisbon, Portugal
daniel.mcnamee@research.fchampalimaud.org

## Abstract

Learned control policies often fail when deployed in real-world environments with changing dynamics. When system dynamics shift unexpectedly, performance degrades until models are retrained on new data. We introduce Reflexive World Models (RWM), a dual control framework that uses world model predictions as implicit reference trajectories for rapid adaptation. Our method separates the control problem into long-term reward maximization through reinforcement learning and robust motor execution through reward-free rapid control in latent space. This dual architecture achieves significantly faster adaptation with low online computational cost compared to model-based RL baselines, while maintaining near-optimal performance. The approach combines the benefits of flexible policy learning through reinforcement learning with rapid error correction capabilities, providing a theoretically grounded method for maintaining performance in high-dimensional continuous control tasks under varying dynamics.

## 1 Introduction

Model-based reinforcement learning has significantly advanced continuous control by integrating learned world models with policy optimization [Hafner et al., 2021, Hansen et al., 2022]. These methods use neural networks to predict future states, enabling both efficient planning through trajectory sampling and stable policy improvement through value estimation. Yet the challenge of maintaining performance under unexpected dynamic shifts—whether from environmental variation, physical wear, or other sources—remains critical. When dynamics change, planning and value computation degrade, often necessitating costly retraining or specific online adaptation mechanisms [Peng et al., 2018, Kumar et al., 2021].

Control theory provides robust and formally grounded methods for handling changing dynamics through adaptive control, offering stability guarantees through Lyapunov analysis [Slotine and Li, 1991]. These methods maintain performance by continuously adjusting control parameters based on tracking errors between desired and actual trajectories. However, classical control approaches rely on explicit reference trajectories and engineered cost functions, limiting their application to problems with well-defined objectives and structured dynamics models [Narendra and Annaswamy, 2012]. This contrasts with reinforcement learning's ability to learn flexible policies from abstract rewards and high-dimensional observations [Sutton and Barto, 2018, Recht, 2019].

39th Conference on Neural Information Processing Systems (NeurIPS 2025).

We present Reflexive World Models (RWM), a framework that repurposes world model predictions as dynamic control targets for rapid adaptation while preserving learned policy behavior. A reinforcement learning module determines optimal trajectories in latent space, which the world model predicts forward in time to serve as references for a control module that maintains performance through trajectory tracking. This architecture is formalized through analysis of value functions, showing how they decompose into slow learning and trajectory stabilization components. Our approach provides a novel mechanism for rapid adaptation by transforming world model predictions into reference trajectories, enabling learned policies to maintain performance under changing dynamics without requiring reward signals, specific robustness procedures, or architectural constraints. In continuous control tasks including locomotion under varying dynamics, this achieves significantly faster adaptation than standard methods while maintaining performance. This computational efficiency arises from RWM's online mechanism, which uses lightweight controller updates driven by latent predictions from the world model. Such direct adjustments bypass the substantial per-step operational costs associated with planning rollouts or the extensive model retraining common in other adaptive strategies.

RWM offers a bridge between classical adaptive control (such as Model Reference Adaptive Control, or MRAC) and modern reinforcement learning. While classical methods provide stability guarantees, their common reliance on pre-defined reference models and state representations can limit their application, especially in complex systems where these are difficult to hand-engineer. RWM addresses this by leveraging a *learned world model* to implicitly derive reference dynamics from high-dimensional observations. Furthermore, its operation in a *learned latent space* allows adaptation based on task-relevant features discovered by the RL agent, rather than fixed, pre-specified ones. By thus deriving references directly from world model predictions, our approach aims to combine the robustness of adaptive control with the flexibility of learned policies, with the value function decomposition establishing performance bounds under varying dynamics. This two-level approach—slower policy learning to discover optimal behaviors and rapid error correction by the adaptive controller to maintain execution fidelity—naturally aligns with hierarchical control architectures. Our experiments demonstrate this separation of policy learning and rapid error correction leads to more efficient adaptation to dynamic changes compared to approaches where these distinct functionalities are not explicitly modularized and managed by dedicated components.

## 2   Background

Modern model-based reinforcement learning integrates several components through learned world models. Given observations of system state, these models learn compressed latent representations where planning and control occur [Hafner et al., 2021]. TD-MPC2 exemplifies this approach through a normalized latent space that enables stable trajectory sampling and value estimation [Hansen et al., 2023]. Previous work has explored different approaches to adaptation - Deep Model Reference Adaptive Control [Joshi et al., 2019] combined neural networks with MRAC but required complex dual architectures, while Rapid Motor Adaptation [Kumar et al., 2021] and Residual Policy Learning [Silver et al., 2019] demonstrated online adaptation but required either specific architectural choices or limited adaptation to particular types of system changes.

Adaptive control provides formal stability guarantees through Lyapunov analysis [Slotine and Li, 1991], achieving millisecond-scale adaptation by adjusting parameters based on trajectory tracking errors. However, these methods require explicit reference trajectories, structured dynamics models, and engineered cost functions [Narendra and Annaswamy, 2012]. In contrast, reinforcement learning learns flexible policies directly from rewards and high-dimensional observations [Sutton and Barto, 2018], but sacrifices adaptation speed and stability guarantees. Recent work on meta-learning [Finn et al., 2017] and domain randomization [Tobin et al., 2017, Tan et al., 2018, Peng et al., 2018] has improved robustness but still requires extensive offline training.

Previous attempts to bridge these approaches have either restricted policies to specific forms amenable to control theory or limited adaptation to particular types of system changes [Recht, 2019]. A general framework for combining the flexibility of learned models with the rapid adaptation of control theory has remained elusive. Our work addresses this gap by leveraging predictive outputs to guide a low-level controller, enabling classical control techniques while maintaining the benefits of learned policies.

# 3 Model design: implicit latent trajectory for adaptive control

## 3.1 Problem Formulation

We consider a continuous control problem formulated as a Markov Decision Process (MDP), defined by the tuple $(S, A, P, R, \gamma)$, where $S$ is the state space, $A$ is the action space, $P(s'|s, a)$ is the transition probability, $R(s, a)$ is the reward function, and $\gamma$ is the discount factor. Our goal is to learn a policy $\pi(a|s)$ that maximizes the expected cumulative discounted reward. We operate in a setting where the true system dynamics, represented by the transition function $P$, can change unexpectedly from the training conditions.

Our approach leverages a learned world model operating in a latent space $Z$. The model consists of: an encoder $\phi : S \to Z$ where $z_t = \phi(s_t)$, a latent dynamics model $F : Z \times A \to Z$ where $\hat{z}_{t+1} = F(z_t, a_t)$, and a base policy $\pi_0 : Z \to A$ where $a_0 = \pi_0(z_t)$. The objective of our Reflexive World Model (RWM) is to learn a second, adaptive control policy $\pi_c : Z \to A$, which outputs a corrective action $a_c$. The total action is $a_t = a_0 + a_c$. The objective for $\pi_c$ is to minimize the discrepancy between the world model's one-step prediction (under the base policy) and the observed outcome, thereby making the system behave as predicted by the reference model under the base policy.

Consider a continuous control MDP with learned policy $\pi_0$ operating through a world model in latent space $z = \phi(s)$. We assume that the latent space captures task-relevant dynamics from observations $s$, with $V(z) = V(\phi(s))$. A world model $F$ predicts future latent states conditioned on the current state and policy actions.

## 3.2 Decomposition of the value objective

Locomotion involves fundamentally distinct learning processes operating at different timescales. Policy learning gradually discovers behaviors that maximize long-term reward. This process requires extensive exploration but develops robust policies for diverse tasks. In contrast, rapid adaptation maintains performance under changing dynamics without modifying the underlying policy. This process responds quickly to errors but operates within the framework of existing behaviors.

This functional separation suggests decomposing motor learning into complementary objectives. The policy learning system should discover behaviors that maximize expected value across tasks, while the adaptation system should maintain stable execution under perturbations. We formalize this approach by linking value-based learning with rapid error correction, decomposing the Taylor expansion of the value function around optimal trajectories in a task-relevant latent space $z$:

$$V(z_{t+1} + \Delta z) = V(z_{t+1}) - \frac{1}{2}\Delta z^T H \Delta z + O(\|\Delta z\|^3) \tag{1}$$

where $H = \nabla^2 V(z_{t+1})$ is negative definite near the optimum. This decomposition suggests separating the problem into maximizing mean value through policy optimization and minimizing deviations through rapid adaptation.

## 3.3 Forward model predictions as references

Our Reflexive World Models (RWM) framework implements the functional separation through a dual architecture. The reinforcement learning module learns a base policy, $a_0 = \pi_0(z)$, that optimizes the mean value, as in classic RL models. We operate within a continuous, normalized latent space $z = \phi(s)$, where $\phi$ is an encoder (further details in Section 5 and Appendix A). We opt for a continuous latent space, akin to TD-MPC2 [Hansen et al., 2023], as it directly supports the differentiability required for our gradient-based adaptive controller (Section 3.3). This choice offers a simpler pathway for encoding states for continuous control tasks compared to discrete representations (e.g., [Hafner et al., 2021]) which would necessitate specialized techniques for gradient propagation. Normalization ensures that all dimensions of the latent space have comparable scales, which is crucial for a balanced contribution to the error computation discussed below. We maintain a forward model $F$ predicting future latent states:

$$\hat{z}_{t+1} = F(z_t, \pi_0(z_t)) \tag{2}$$

The world model $F$ and the policy $\pi_0$ (and by extension, the encoder $\phi$) are assumed to be differentiable with respect to their inputs. This is a common assumption in many model-based RL approaches

that use gradient-based learning and is a prerequisite for our adaptive control gradient computation (Section 3.3).

Our framework builds on model-based reinforcement learning but changes how world model predictions drive behavior. A conceptual novelty is that we interpret the forward model predictions as target states. Both the forward model and controller share the same error function measuring discrepancy between predicted and actual states:

$$\mathcal{L} = \|\hat{z}_{t+1} - z_{t+1}\|^2 \tag{3}$$

Following approaches like TD-MPC2 [Hansen et al., 2023], we use the Mean Squared Error (MSE) as the discrepancy measure. Given the normalized latent space, MSE provides a well-distributed measure of prediction error across all latent dimensions. This choice contrasts with scale-invariant metrics such as KL-divergence, utilized in some world models (e.g., [Hafner et al., 2021]); we found empirically that KL-divergence yields worse adaptation performance in our experiments. A scale-invariant loss can disproportionately weight or neglect certain latent dimensions whose precise tracking might be critical for fine-grained motor adaptation. However, the forward model and control module minimize this error in opposing ways. The forward model adapts its predictions to match observations, following the standard gradient to improve predictions. In contrast, the control module adapts actions to make the system behave as predicted.

### 3.4 Adaptive control gradients

This approach inverts the standard relationship between models and control. Classical adaptive control assumes reference trajectories and adapts a controller to track them. Model-based RL learns models that predict actual outcomes and uses them for planning. Our framework generates reference trajectories directly from world model predictions while adapting control to maintain their validity under changing dynamics.

The control policy updates follow a modified gradient computation that reflects this inversion. Rather than updating predictions to match observations, we update actions to make observations match predictions:

$$\theta_c \leftarrow \theta_c - \eta_c \left( -\frac{\partial \mathcal{L}}{\partial a_0} \right) \left( \frac{\partial a_c}{\partial \theta_c} \right) \tag{4}$$

The update function leverages gradients through the world model to determine how actions should change to reduce prediction error, inverting the standard approach to model learning. This differs fundamentally from standard practice where gradients flow from predictions to parameters. The control module instead treats predictions as fixed targets and adapts actions to achieve them.

The total action combines the base policy with these corrections:

$$a_t = \pi_0(z_t) + \pi_c(z_t) \tag{5}$$

Critically, this update requires only prediction error—no reward signal is needed for adaptation. Operating in the world model's latent space provides two benefits. First, it ensures the control module focuses adaptation on task-relevant features captured by the learned representation. Second, it provides an interface between the RL policy operating on compressed latent states and the control module maintaining prediction consistency.

## 4 Theoretical Guarantees

A notable feature of the RWM framework is that this decoupling of long-term policy optimization from rapid, reference-tracking adaptive control makes the system particularly amenable to rigorous control-theoretic analysis. For many contemporary model-based RL agents (e.g., [Hafner et al., 2021, Hansen et al., 2022]), deriving formal guarantees on overall task reward or value under dynamic perturbations is exceptionally challenging. This difficulty often arises not just from the inherent complexity of these comprehensive, end-to-end learned systems. It also stems from the fact that the general, high-dimensional value functions they optimize are not easily subjected to direct stability or error-bound analysis from classical control theory.

In contrast, RWM's specific formulation of the adaptive control objective—minimizing the latent prediction error $\|\hat{z}_{t+1} - z_{t+1}\|^2$ (Equation 3) to ensure the system tracks the world model's predictions—presents a more constrained and tractable problem. This focus on a well-defined error signal

**Algorithm 1** Reflexive World Models (RWM)
---
**Require:** Trained policy $\pi_0$, encoder $\phi$, world model $F$, learning rate $\eta_c$
**Ensure:** Adapted control policy $\pi_c$
  Initialize $\pi_c$
  **for** each episode **do**
    $z_t \leftarrow \phi(s_t)$
    $a_0 \leftarrow \pi_0(z_t); a_c \leftarrow \pi_c(z_t)$
    Execute $a_t = a_0 + a_c$, observe $s_{t+1}$
    $z_{t+1} \leftarrow \phi(s_{t+1})$
    $\hat{z}_{t+1} \leftarrow F(z_t, a_0)$
    $e_t \leftarrow z_{t+1} - \hat{z}_{t+1}$
    $\theta_c \leftarrow \theta_c - \eta_c \left( -\frac{\partial \|e_t\|^2}{\partial a_0} \right) \frac{\partial a_c}{\partial \theta_c}$
  **end for**
---

for the adaptive controller allows us to apply established control-theoretic tools. Consequently, we can establish theoretical limits on this control error and, crucially, link these to bounds on the overall value function (Theorem 4.3). This provides formal insights into expected performance degradation under changing dynamics, bridging a gap between flexible learning and provable robustness.

The following theorems formalize these guarantees:

**Assumption 4.1** (System Properties). *The system satisfies:*

1. $\|\partial F/\partial a\| \leq L$                *(Lipschitz control)*

2. $\sigma_{min}(\partial F/\partial a) \geq \alpha > LP$      *(control authority)*

3. $\|F(z,a) - f(z,a)\| \leq \epsilon$          *(model accuracy)*

4. $\|p(t)\| \leq P$         *(bounded perturbation)*

*where $P$ bounds external perturbations.*

**Theorem 4.2** (Control Error). *Under Assumption 4.1, the control law $a_c = -\eta(\partial F/\partial a)^T e(t)$ achieves:*

$$\|e(t)\| \leq \gamma^t \|e(0)\| + \sqrt{\epsilon^2 + \frac{P^2}{\alpha^2}} \tag{6}$$

*where $\gamma = (1 - \eta\alpha^2 + \eta L^2) < 1$ for $\eta < 1/L^2$.*

**Theorem 4.3** (Value Bounds). *If the error bound is sufficiently small, the value function satisfies:*

$$V(z^*) - V(z) \leq \frac{H_M}{2} \left( \epsilon^2 + \frac{P^2}{\alpha^2} \right) \tag{7}$$

*where $H_M$ bounds the eigenvalues of $-\nabla^2 V$ near optimal trajectories.*

These results show how world model accuracy ($\epsilon$), control authority ($\alpha$), and perturbation magnitude ($P$) determine performance bounds, with quadratic scaling reflecting the natural structure of value functions around optimal trajectories. The proofs use standard Lyapunov techniques (see Appendix F).

## 5 Simulations

Our experiments address three questions: (1) Can world model predictions serve as effective reference trajectories for rapid adaptation? (2) Does minimizing control error—without access to reward—achieve robust adaptation across diverse tasks? (3) How does performance scale from simple to high-dimensional systems?

We evaluate across multiple continuous control environments from the DeepMind Control Suite [Tassa et al., 2018]: a 2D point-mass system (direct state observations, no encoder), standard locomotion (Walker-Run, Cheetah-Run, Hopper-Hop), manipulation (Ball-in-Cup), and high-dimensional coordination (Humanoid-Walk, 17 actuators). Adaptation is tested under actuator perturbations simulating online miscalibration, comparing RWM against frozen baseline policies and reward-based fine-tuning.

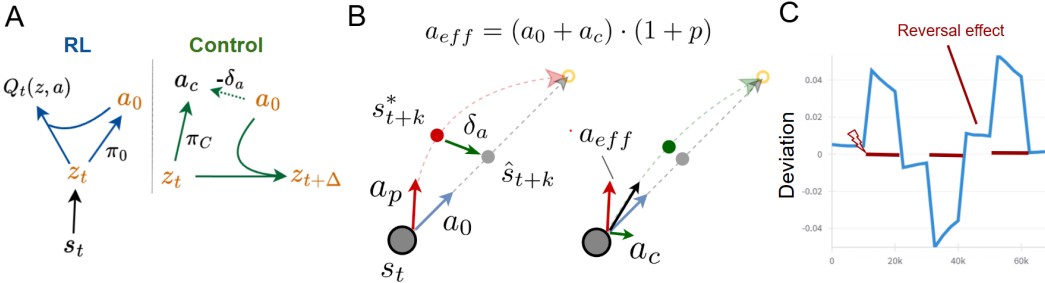

Figure 1: (A) Network architecture of Reflexive World Models (RWM), showing the reinforcement learning policy (blue) and adaptive control modules (green), with interface variables in orange. Each transformation is implemented as a two-hidden-layer MLP. (B) Illustrative simulation of the adaptive control mechanism for a 2D pointmass task (without encoder, $z = s$). When actuators are perturbed, the trajectory deviates from the predicted future states $\hat{s}_{t+k}$ under the base policy actions $a_0$. This error triggers an update to generate corrective actions $a_c$. (C) Under alternating directional perturbations (red), RWM corrects deviations from the optimal trajectory, exhibiting characteristic after-effects when perturbations are removed.

## 5.1 Addressing Policy Saturation and Inaction via a Thresholded Action Cost

Our adaptive controller $\pi_c$ requires the base policy $\pi_0$ to provide differentiable, non-saturated actions. However, standard RL policies in continuous control can exhibit action saturation (leading to vanishing gradients for $\pi_c$, see Fig. 4-C) or policy inaction (the "dead problem" from naive action costs, Fig. 4-A,B). Both issues hinder effective adaptation.

To ensure a suitable base policy, we incorporate a specific thresholded quadratic action cost, $\lambda \sum_i (\max(0, |a_i| - c))^2$, during $\pi_0$'s pre-training. This simple modification to standard action penalization encourages $\pi_0$ to operate within a smoother, non-saturated region responsive to $\pi_c$, while also mitigating policy inaction. The formulation, rationale, and illustrative effects (Fig. 4-D) are detailed in Appendix C.

## 5.2 Dual-model Design

Our Reflexive World Models (RWM) approach builds on model-based reinforcement learning methods such as Dreamer and TD-MPC2 [Hafner et al., 2021, Hansen et al., 2023], primarily to leverage their task-related encoders. In particular, TD-MPC2 uses latent, reward and value predictions to learn the encoder and forward models. We reuse these pre-trained representations to focus on the novel control mechanism. In simple environments where an encoder is unnecessary, such as the point-mass task, we use direct state observations ($z_t = s_t$). In these cases, SAC or any alternative policy, including non-RL-based controllers, could be used as the baseline policy, and the forward model is learned through latent prediction.

## 5.3 Perturbation Experiments

We evaluate adaptation by introducing perturbations $p(t)$ in the action space, with an effective actuator $a_{eff} = (a_0 + a_c) \cdot (1 + p)$, and measuring the controller's ability to compensate for them (Fig. 1-B). Step perturbations involve sudden changes in the action signal at specific intervals, while slow perturbations introduce gradual, non-stationary shifts that mimic actuator miscalibration. These perturbations allow us to analyze how the controller reacts to both abrupt and progressive deviations.

Our primary baseline uses the TD-MPC2 reactive policy trained without the online planner, with continued online fine-tuning. This configuration was chosen to avoid conflating the use of the world model for both planning and our control mechanism, ensuring a cleaner experimental comparison. The original TD-MPC2 work [Hansen et al., 2023] demonstrated that the online planner provides substantial benefits primarily on complex tasks, while the reactive policy remains highly performant on standard locomotion—a finding we confirm empirically in Appendix E, where both configurations achieve nearly identical performance on Walker, Cheetah, and Hopper tasks. The exception is the

Humanoid environment, where the reactive policy alone is insufficient for stable locomotion, so we use TD-MPC2 with its planner for this task only.

The experiment consists of two phases. In the first phase, a policy is trained or provided without perturbations, and a forward model is learned using its trajectories. This phase establishes the baseline model of the system's behavior under normal conditions. In the second phase, perturbations are introduced while simultaneously activating the controller. The controller uses the learned forward model to adjust its outputs in response to the deviations introduced by the perturbations.

To assess adaptation performance, we measure the drop in task performance caused by the perturbations and track the forward model error. The latter serves as an implicit measure of latent trajectory deviation, reflecting how well the system follows the predictions of the forward model.

The prerequisite pre-training baseline model is common to all and takes approximately 16-17 hours. Execution times for the experimental setups are detailed in Table 1. RWM's online phase, which uses pre-trained components and lightweight controller updates, is notably more computationally efficient than approaches requiring full model retraining.

Table 1: Typical execution times for the online adaptation phase (1 million environment steps under perturbation) on a single NVIDIA Tesla T4 GPU.

| Experimental Setup | Approximate Execution Time |
| --- | --- |
| TD-MPC2 (Full Training) | 16.5 hours |
| RWM (Online Adaptation) | 1.8 hours |
| No Adaptation (Inference) | 1.4 hours |

The computational efficiency of RWM's online adaptation (Table 1) stems from using lightweight controller updates based on forward-model predictions, avoiding the computational overhead of planning horizons or full model retraining typical in model-based approaches.

### 5.4 Correction for Trajectory Deviations

The point mass system demonstrates the core interaction between policy and adaptation modules. Under angular perturbation, the base policy's trajectories systematically deviate from target while the world model maintains predictions of intended paths (Fig. 1-B). The control module uses these predictions as references to generate corrective actions, recovering performance without modifying the underlying policy.

The system exhibits characteristic aftereffects when perturbations are removed (Fig. 1-C). Initial overcorrection in the opposite direction indicates adaptation through internal model formation rather than reactive control. This validates the method's ability to learn and compensate for systematic changes in dynamics while preserving the original policy.

### 5.5 Robust Motor Control

The Walker2D environment demonstrates how adaptation can operate effectively in learned latent space. Under step perturbations to actuator gains, the control module rapidly reduces the error between predicted and actual latent states (Fig. 2). The right column of Figure 2 presents these metrics aggregated over perturbation cycles, with values normalized within each cycle relative to the 'No Adaptation' agent's performance range to highlight relative improvements. As the latent prediction error decreases (shown in the control error plots), task performance improves correspondingly (shown in the reward plots), validating that world model predictions in latent space provide effective references for adaptation.

Table 2 presents results across multiple environments under step perturbations (full results in Appendix B). Critically, RWM achieves robust adaptation without access to reward signals—minimizing latent control error alone is sufficient to recover substantial task performance. On Walker, RWM achieves 94.3% reward recovery, substantially outperforming fine-tuning's 66.3%, demonstrating that the reward-free objective can match or exceed reward-driven adaptation. The Humanoid environment is particularly revealing for complex, high-dimensional tasks: RWM's controller successfully operates on the 17-actuator system, while fine-tuning on reward causes the policy to deviate significantly from

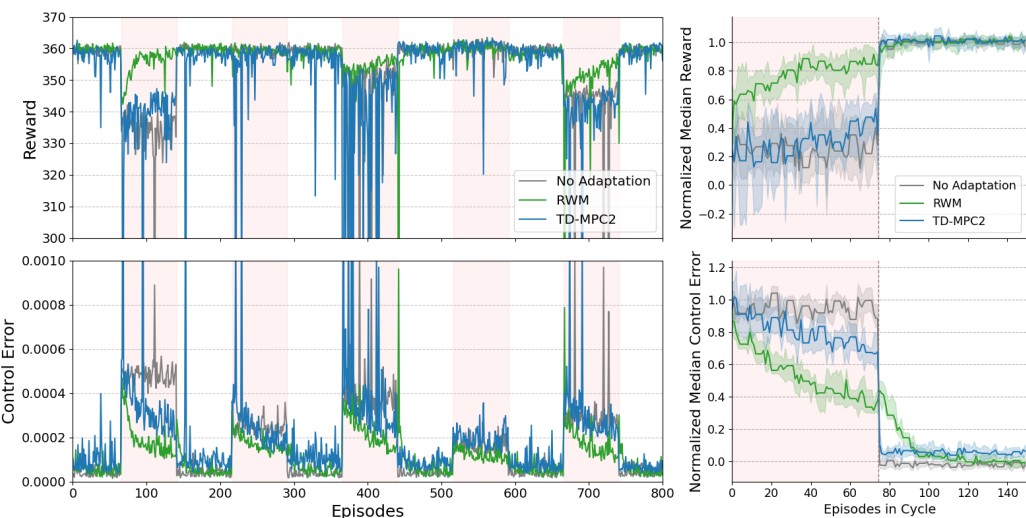

Figure 2: RWM adaptation performance under step motor perturbations. The plots show Reward and Control Error over 800 episodes (left column); shaded areas indicate perturbation periods. The right column displays Normalized Median Reward and Control Error, aggregated across perturbation cycles. Shaded areas in the right plots represent the 95% confidence interval of the median (bootstrapped). RWM (green line) consistently maintains higher reward and lower control error compared to No Adaptation (gray line) and the TD-MPC2 (no planner) baseline (blue line), demonstrating effective and rapid recovery from perturbations.

its original motor program (58.6% increase in control error). In contrast, RWM preserves the core motor pattern (24.1% decrease in control error). This illustrates a central distinction—reward reflects narrow task-specific metrics (e.g., forward velocity), while control error captures fidelity to the entire learned motor program. In complex tasks, these objectives can diverge: maximizing reward may degrade the underlying coordination pattern, suggesting that preserving trajectory integrity through control error minimization can provide a more robust foundation for adaptation.

Table 2: Adaptation performance on representative environments under step perturbations. Reward Drop and Control Error measure degradation from baseline; Improvement (%) shows recovery relative to frozen policy. See Appendix B for complete results across all environments.

| Environment | Method | Reward Drop | Control Error | Improvement (%) Reward | Control |
|---|---|---|---|---|---|
| Walker-Run | Frozen Policy | 108.40 | 28.09 | – | – |
| | Fine-tuning | 36.50 | 8.87 | 66.3 | 68.4 |
| | RWM (ours) | 6.20 | 7.43 | **94.3** | **73.6** |
| Humanoid-Walk | Frozen Policy (w/ planner) | 28.10 | 0.58 | – | – |
| | Fine-tuning (w/ planner) | 12.90 | 0.92 | **54.1** | -58.6 |
| | RWM (ours, w/ planner) | 13.60 | 0.44 | 51.6 | **24.1** |

A crucial test for deployment in real-world systems is the ability to handle nonstationary dynamics—scenarios where system properties gradually change over time due to wear, temperature fluctuations, or miscalibration. Such scenarios are particularly challenging because they cannot be addressed through fixed robustness strategies, demanding continuous adaptation. To evaluate this capability, we introduced continuously varying perturbations by applying filtered noise to actuator gains, simulating the gradual deterioration and drift common in physical hardware (Fig. 3A). The results suggest strong potential for real-world applications: RWM maintains a performance of 360.56, substantially outperforming both TD-MPC2 (no planner) (311.67) and the fixed baseline policy which degrades to 233.42. This performance advantage is accompanied by systematically lower control error for

RWM compared to both alternatives, indicating more efficient adaptation to continuously changing dynamics. The world model predictions provide a stable reference for ongoing adaptation even as dynamics evolve unpredictably, enabling rapid corrections without reward signals or policy retraining that would be impractical in deployed systems.

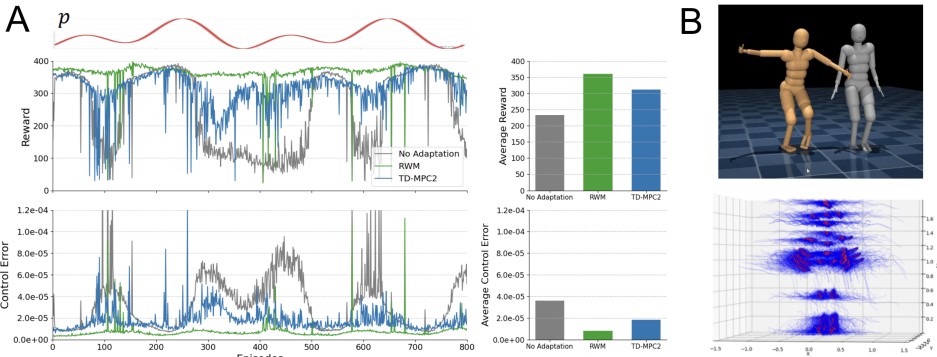

Figure 3: Nonstationary perturbations and high-dimensional coordination. (A) Walker2D under continuous filtered noise perturbations to actuator gains, following sinusoidal pattern $p$ (top). Time series (left) and averaged performance (right) show RWM (green) achieves the highest reward (360.56), followed by TD-MPC2 (no planner) (blue) (311.67), with No Adaptation (gray) performing worst (233.42). Control error measurements (bottom) demonstrate that RWM maintains systematically lower error throughout adaptation compared to both alternatives. (B) Analysis of the 17-actuator Humanoid environment showing coordinated movement patterns maintained by RWM even under perturbations.

## 5.6 High-Dimensional Coordination and a Control-Theoretic Perspective

Beyond nonstationary dynamics, we examined high-dimensional coordination in the 17-actuator Humanoid environment (Fig. 3B), where successful locomotion requires synchronized movements across multiple joints. When perturbed, the unadapted system exhibits severely degraded coordination between joints (compare the normal orange gait with the gray affected movement). RWM's primary advantage is revealed in the trajectory density visualization: by tracking latent predictions that capture coordinated multi-joint relationships, it maintains coherent movement patterns even under challenging conditions.

This separation of concerns between RL and control modules provides a framework to analyze different sources of variability: intentional variability from exploration and decision-making (RL module) versus unwanted variability from imperfect execution and external perturbations (control module). This distinction has potential applications in understanding biological motor control, where similar separations between voluntary movements and reflexive corrections exist. The system demonstrates sophisticated adaptation where perturbations to individual actuators trigger compensatory adjustments across multiple joints simultaneously, preserving overall balance and performance.

## 5.7 Comparison with Domain-Randomized Baselines

To isolate the benefits of RWM's online adaptive mechanism, we compared its performance against baseline agents (TD-MPC2 (no planner) and No Adaptation) that were pre-trained with exposure to the same randomized actuator perturbations subsequently used during evaluation. This form of domain randomization during pre-training aims to enhance the inherent robustness of the baseline policies. The comparison then assesses whether RWM's dedicated online adaptation still provides substantial advantages beyond this enhanced baseline robustness. The results of this comparison are presented in Figure 5 (see Appendix D).

This comparison investigates whether pre-training the baseline TD-MPC2 (no planner) policy with exposure to randomized actuator perturbations—a form of domain randomization—could achieve robustness comparable to RWM's online adaptation. While such pre-training does enhance the baseline's performance when dynamics are nominal (i.e., perturbation OFF in Figure 5 (see Appendix D),

where this specialized TD-MPC2 (no planner) agent achieves higher reward than RWM), the critical test is performance under active, unmodelled dynamic shifts. During these perturbation phases, RWM's advantage is evident: Table 3 shows that RWM maintains significantly higher reward and lower control error compared to the TD-MPC2 (no planner) agent pre-trained with perturbations. This outcome underscores that while domain randomization can improve baseline robustness to some extent (as the pre-trained No Adaptation policy still performed worst, with a median reward of 0.1150 and control error of 0.9676 during perturbation), it does not eliminate the performance degradation caused by the perturbations encountered during evaluation. Consequently, this form of domain randomization does not eliminate the need for an explicit online adaptation mechanism like RWM, which provides more effective real-time compensation for unmodelled dynamics than pre-training alone.

Table 3: Comparison with Domain-Randomized Baselines: Performance Metrics during Perturbation ON (Baselines Pre-trained with Perturbations)

| Metric | RWM | TD-MPC2 (no planner) |
|---|---|---|
| Median Normalized Reward | 0.6003 | 0.3600 |
| Median Normalized Control Error | 0.7307 | 1.4162 |

## 6 Discussion

This work introduces Reflexive World Models (RWM), demonstrating how predictive models can be cast as sources of desired state trajectories for rapid adaptation. RWM distinctively inverts the standard model-based RL paradigm: rather than using world models for planning, we enable the control policy $\pi_c$ to directly minimize prediction errors in latent space. This mechanism reframes the learning objective—instead of the model learning to track reality, the agent acts to make reality track the model. This contrasts with typical model-based RL methods where world models primarily serve for planning or data augmentation. Unlike Model Predictive Control, which plans action sequences over a horizon, RWM uses single-step predictions as instantaneous references, enabling rapid responses with low computational overhead ideal for real-world deployment. A notable advantage of this approach is that adaptation requires no reward signal, making it applicable when rewards are sparse, delayed, or unavailable in deployed systems.

The inherent dual-timescale operation in RWM—slower policy learning with $\pi_0$ discovering optimal behaviors and faster error correction via $\pi_c$ maintaining execution fidelity—is a central aspect of its effectiveness. This separation of concerns, where $\pi_c$ rapidly compensates for dynamic shifts based on $\pi_0$'s intended trajectory (as predicted by the world model), allows the system to adapt without costly retraining of the entire policy or world model. These theoretical guarantees (Section 4) rely on assumptions such as sufficient control authority and model accuracy, which may not always be fully met in highly complex or underactuated scenarios. Nevertheless, our empirical results demonstrate RWM's practical effectiveness across challenging benchmarks.

### 6.1 Limitations and Future Work

Future work includes several directions. First, learning latent representations specifically optimized for RWM's adaptive control could enable application to arbitrary pre-trained policies. Second, while our phased training approach simplifies analysis, end-to-end joint learning of the base policy, world model, and adaptive controller presents interesting challenges around preventing controller exploitation of model inaccuracies. Third, extending RWM beyond actuator perturbations to handle morphological changes, environmental shifts, or sensor noise would broaden applicability. This may require incorporating uncertainty estimation into the world model or developing methods to disambiguate different error sources.

The principle of reward-free adaptation through world model predictions offers a promising foundation for robust, learned behaviors in real-world deployments where reward signals may be unavailable. Reflexive World Models (RWM) build on this by integrating a rigorous theoretical framework with comprehensive empirical support, facilitating the development of adaptive agents that can sustain high performance under real-world dynamic changes.

## Acknowledgments and Disclosure of Funding

This work was supported by the Champalimaud Foundation and the Google Cloud Research Credits program with the award GCP398030901.

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

# A  Implementation Details

Our main simulations are conducted using the DeepMind Control Suite [Tassa et al., 2018], with tasks such as `humanoid-walk` and `walker-walk` (with an action repeat of 2), and are implemented in PyTorch [Paszke et al., 2019]. Experiments were run on a computer with a single NVIDIA Tesla T4 GPU (16 GB GDDR6). For experimental runs of 1 million environment steps, typical execution times on this GPU were approximately 1.5-2 hours for the 'No Adaptation' and RWM setups. These setups primarily involve inference using pre-trained components, with RWM additionally performing a lightweight online update for its controller. In contrast, a full TD-MPC2 training run of 1 million steps, which includes its comprehensive learning cycle of model updates, planning, and policy/value optimization, typically took around 16-17 hours.

**Baseline TD-MPC2 Agent.** We pre-train a TD-MPC2 agent [Hansen et al., 2023] for 1 million environment steps to provide the state encoder $\phi$, the base policy $\pi_0$, and the forward dynamics model $F$. The encoder $\phi$ consists of 2 hidden layers with 256 units each, while other MLPs within the TD-MPC2 agent (such as those for the policy and dynamics model) have 2 hidden layers with 512 units each. These networks use Mish activations. The encoder output is normalized using SimNorm with 8 dimensions [Hansen et al., 2023]. The base policy $\pi_0$ is pre-trained with a thresholded quadratic action cost (Section 5.1, with full details and illustration in Appendix C). The environments use hard action bounds of $[-2, 2]$, and the threshold $c$ for this cost is set to 0.5, with the penalty coefficient $\lambda = 0.2$.

**Forward Model Training.** The forward model $F$ is trained as a general transition model during the pre-training phase, exposed to a wide distribution of states and actions including those from early, suboptimal policies—not just trajectories from the final converged policy. This broad training enables the model to make reasonable predictions even for states off the optimal trajectory, which is crucial for providing corrective signals when perturbations occur.

**Episode Length and Reward Scale.** Our experiments use a maximum episode length of 400 steps, compared to 1000 steps in the original TD-MPC2 paper [Hansen et al., 2023]. This results in maximum achievable rewards of approximately 400 for locomotion tasks, rather than 800+. Our reported rewards of $\sim$380 are therefore near-optimal for this episode length. The thresholded action cost is applied only during policy training to ensure sufficient action headroom for the adaptive controller.

For comprehensive details on the TD-MPC2 architecture and its training, we refer to the original publication [Hansen et al., 2023].

**RWM Controller.** The Reflexive World Model controller, $\pi_c$, is an MLP with 2 hidden layers and 512 units per layer, using ReLU activations. During RWM adaptation, the parameters of the pre-trained TD-MPC2 components (encoder $\phi$, base policy $\pi_0$, and forward model $F$) are frozen. The RWM controller $\pi_c$ is then trained online over a multi-step horizon of 3 to minimize the squared error between the forward model's latent state predictions (based on $\pi_0$'s actions) and the observed next latent states, as per Algorithm 1. This update involves backpropagating the gradient of this prediction error with respect to the base policy action $a_0$, and then transferring this gradient (with an inverted sign, as detailed in Algorithm 1 and Equation 4) to update the RWM controller $\pi_c$. The learning rate for $\pi_c$ is $3 \times 10^{-4}$.

**Perturbation Protocol.** To evaluate adaptation, we introduce multiplicative step perturbations to actuator gains, where the effective action for each actuator $j$ becomes $a_{eff,j} = (a_{0,j} + a_{c,j}) \cdot (1 + p_j)$. For each perturbation instance, the value $p_j$ for each actuator $j$ is sampled uniformly from the range $[-0.5, 0.5]$. These perturbations alter actuator outputs for 10,000 to 15,000 environment steps, followed by an equal duration without perturbation. This cycle repeats throughout an experimental run, which can extend up to 1 million steps for comprehensive experiments including pre-training and adaptation phases.

All neural networks, including $\pi_c$, are optimized using Adam [Kingma and Ba, 2015]. A simplified reference implementation of the RWM controller is provided in the supplementary material. The full codebase will be made available upon publication.

## B    Results Across All Environments

Table 4 presents adaptation performance across all evaluated environments under step perturbations.

Table 4: Adaptation performance across all environments under step perturbations. Reward Drop and Control Error measure degradation from baseline; Improvement (%) shows recovery relative to frozen policy. RWM achieves adaptation through reward-free control error minimization.

| Environment | Method | Reward Drop | Control Error | Improvement (%) | |
| --- | --- | --- | --- | --- | --- |
| | | | | Reward | Control |
| | Frozen Policy | 1.00 | 9.05 | – | – |
| Ball-in-Cup | Fine-tuning | 0.00 | 2.61 | **100.0** | 71.2 |
| | RWM (ours) | 1.00 | 2.09 | 0.0 | **77.0** |
| | Frozen Policy | 6.30 | 3.21 | – | – |
| Cheetah-Run | Fine-tuning | 5.60 | 2.44 | 11.1 | 24.0 |
| | RWM (ours) | 1.40 | 1.70 | **77.8** | **47.0** |
| | Frozen Policy | 15.40 | 5.67 | – | – |
| Hopper-Hop | Fine-tuning | 6.20 | 2.92 | 59.7 | **48.5** |
| | RWM (ours) | 4.50 | 3.20 | **70.8** | 43.6 |
| | Frozen Policy | 108.40 | 28.09 | – | – |
| Walker-Run | Fine-tuning | 36.50 | 8.87 | 66.3 | 68.4 |
| | RWM (ours) | 6.20 | 7.43 | **94.3** | **73.6** |
| | Frozen Policy (w/ planner) | 28.10 | 0.58 | – | – |
| Humanoid-Walk | Fine-tuning (w/ planner) | 12.90 | 0.92 | **54.1** | -58.6 |
| | RWM (ours, w/ planner) | 13.60 | 0.44 | 51.6 | **24.1** |

## C    Details on the Thresholded Action Cost

A base policy $\pi_0$ that provides differentiable, non-saturated actions is crucial for the effectiveness of our adaptive controller, $\pi_c$. Standard continuous control policies, however, can suffer from two common issues: (i) action saturation at the boundaries of the allowed range, which nullifies gradients needed by $\pi_c$ (Fig. 4-C), and (ii) policy inaction or the "dead problem" (Fig. 4-A,B), where naive quadratic action costs can overly penalize any movement, leading to minimal activity.

To address these, we employ a *thresholded quadratic action cost*, $\lambda \sum_i (\max(0, |a_i| - c))^2$, during the pre-training of $\pi_0$. Here, $a_i$ is an action component, $c$ is a threshold (e.g., $c = 0.5$), and $\lambda$ a coefficient (e.g., $\lambda = 0.2$). This cost is a simple modification of a standard quadratic penalty but only penalizes action magnitudes $|a_i|$ exceeding $c$. This encourages $\pi_0$ to operate primarily within a smoother, non-saturated region (Fig. 4-D), preserving differentiability for $\pi_c$. Simultaneously, by not penalizing actions within $[-c, c]$, it mitigates the dead problem, allowing the agent to achieve adequate task performance. While other techniques might yield suitably bounded actions, this approach offers a straightforward way to obtain a responsive baseline for adaptation.

## D    Figure for Comparison with Domain-Randomized Baselines

Figure 5 illustrates the performance comparison for the comparison with domain-randomized baselines discussed in Section 5.7, where baseline policies were pre-trained with exposure to actuator perturbations.

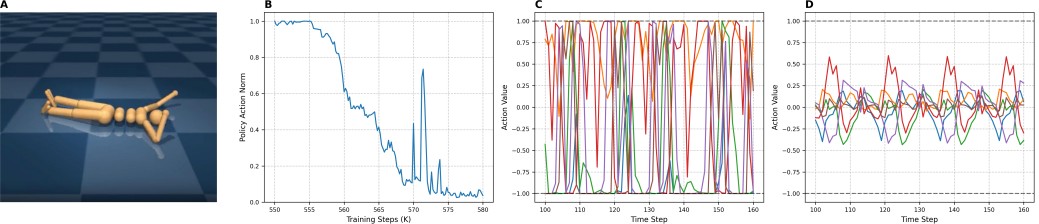

Figure 4: Addressing challenges in baseline policy actions for effective adaptation. (A) A humanoid agent exhibiting "dead" behavior due to a simple quadratic action cost in its RL objective, leading to inaction. (B) The norm of actions for the simulation in (A), demonstrating a decay towards zero over training episodes as the agent minimizes the naive action cost. (C) Action component values over time for a standard TD-MPC2 policy (without the thresholded cost) in the Humanoid task, showing frequent saturation at the boundaries [-1, 1], which impedes gradient flow for the adaptive controller. (D) Smoother and bounded action values from a TD-MPC2 policy trained with the proposed thresholded quadratic action cost, maintaining differentiability and responsiveness.

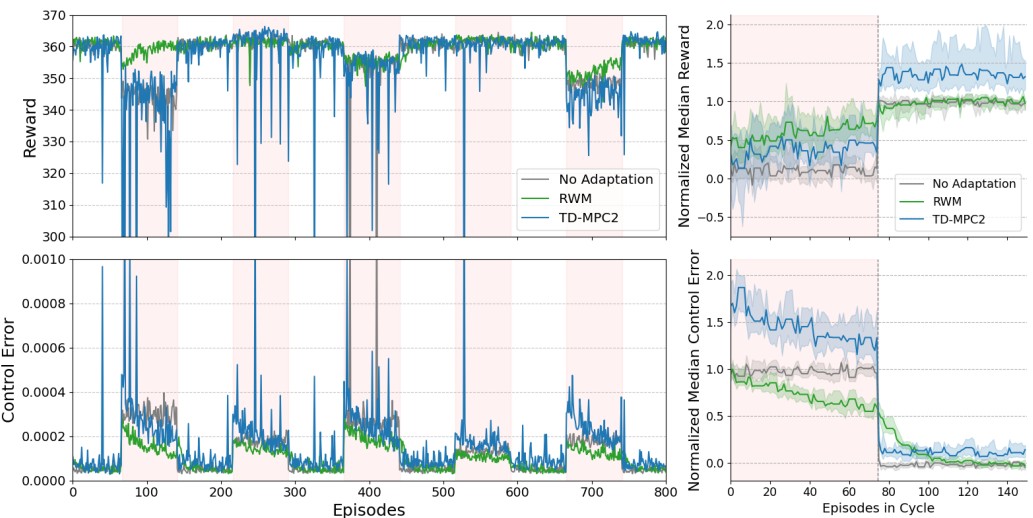

Figure 5: Comparison with Domain-Randomized Baselines: Impact of pre-training with perturbations. Comparison of No Adaptation, RWM, and TD-MPC2 (no planner) when baseline policies for No Adaptation and TD-MPC2 are pre-trained with exposure to actuator perturbations. (Left column) Reward and Control Error over episodes. (Right column) Normalized median reward and control error within perturbation cycles. While pre-training with perturbations improves the baseline, RWM (green line) still demonstrates superior adaptation capabilities in terms of reward and control error compared to the pre-trained TD-MPC2 (no planner) (blue line) and the pre-trained No Adaptation policy (gray line).

# E  Baseline Comparison: TD-MPC2 With and Without Planner

We use the TD-MPC2 reactive policy (trained without the online planner) as our primary baseline. This appendix provides empirical validation that this configuration is appropriate for the locomotion tasks evaluated.

Table 5 shows that for standard locomotion (Walker, Cheetah, Hopper), the reactive policy achieves nearly identical performance to the full agent with planning, both in final reward and when handling perturbations. This is consistent with the original TD-MPC2 work [Hansen et al., 2023], which showed the online planner provides substantial benefits primarily on complex tasks like Humanoid. Importantly, TD-MPC2 without the planner still leverages the world model for latent imagination

during policy learning (similar to Dreamer [Hafner et al., 2021]), making it a capable model-based baseline.

Table 5: TD-MPC2 performance with and without online planner on standard locomotion tasks.

| Environment | Configuration | Final Reward (1M steps) | Reward Under Perturbation |
|---|---|---|---|
| Walker-Run | Without Planner | 380.2 | 342.1 |
| | With Planner | 381.4 | 345.3 |
| Cheetah-Run | Without Planner | 364.5 | 358.2 |
| | With Planner | 362.8 | 360.1 |
| Hopper-Hop | Without Planner | 373.1 | 367.4 |
| | With Planner | 372.6 | 368.2 |

Table 6 compares against SAC, a standard model-free baseline. TD-MPC2 (no planner) substantially outperforms SAC.

Table 6: Comparison of TD-MPC2 (no planner) against model-free SAC baseline.

| Environment | SAC | TD-MPC2 (no planner) |
|---|---|---|
| Cheetah-Run | 244.0 | 364.5 |
| Walker-Run | 249.7 | 380.2 |
| Hopper-Hop | 19.2 | 373.1 |

These results validate our choice: the reactive policy performs comparably to the full agent under perturbation and substantially outperforms model-free methods. This configuration enabled a clean comparison by avoiding conflation of world model use for both planning and our control mechanism.

# F    Theoretical Analysis

We provide proofs for the main theoretical results, showing how control error bounds lead to value function guarantees.

The proofs rely on Assumption 4.1 from Section 4.

**Theorem F.1** (Control Error Bounds). *Under Assumption 4.1, the control law $a_c = -\eta(\partial F/\partial a)^T e(t)$ with $\eta < 1/L^2$ achieves:*

$$\|e(t)\| \leq \gamma^t \|e(0)\| + \sqrt{\epsilon^2 + \frac{P^2}{\alpha^2}} \tag{8}$$

*where $\gamma = (1 - \eta\alpha^2 + \eta L^2) < 1$.*

*Proof.* The error evolves as:

$$e(t+1) = \underbrace{F(z_t, a_t) - F(z_t, a_0)}_{\text{control effect}} + \underbrace{F(z_t, a_0) - f(z_t, a_t)}_{\text{model error}} + p(t) \tag{9}$$

For the control effect:

$$-F(z, a_0 + a_c) = -\eta(\partial F/\partial a)(\partial F/\partial a)^T e(t) \quad \text{(first order)} \tag{10}$$

$$\|(\partial F/\partial a)(\partial F/\partial a)^T\| \geq \alpha^2 \quad \text{(by min singular value)} \tag{11}$$

$$\|F(z, a_0 + a_c) - F(z, a_0)\| \leq (1 - \eta\alpha^2 + \eta L^2)\|e(t)\| \quad \text{(Lipschitz)} \tag{12}$$

The model error satisfies:

$$\|F(z, a_0) - f(z, a_t)\| \leq \epsilon + L\|a_c\| \leq \epsilon + \frac{LP}{\alpha} \tag{13}$$

Therefore:

$$\|e(t+1)\| \leq \gamma\|e(t)\| + \epsilon + \frac{P}{\alpha} \tag{14}$$

By condition (2) of Assumption 4.1 and $\eta < 1/L^2$:

$$\gamma = 1 - \eta\alpha^2 + \eta L^2 < 1 - \eta(LP)^2 + \eta L^2 < 1 \tag{15}$$

The bound follows from solving this recurrence, using the fact that for positive $a, b$:

$$(a + b)^2 \leq 2(a^2 + b^2) \tag{16}$$

$\square$

**Theorem F.2** (Performance Guarantees). *If $\sqrt{\epsilon^2 + P^2/\alpha^2} < \delta$ where $\delta$ bounds the region of quadratic approximation for $V$, then:*

$$V(z^*) - V(z) \leq \frac{H_M}{2}\left(\epsilon^2 + \frac{P^2}{\alpha^2}\right) \tag{17}$$

*where $H_M$ bounds the eigenvalues of $-\nabla^2 V$.*

*Proof.* Around optimal trajectories, Taylor expansion gives:

$$V(z^* + \Delta z) = V(z^*) - \frac{1}{2}\Delta z^T H \Delta z + R(\Delta z) \tag{18}$$

where $|R(\Delta z)| \leq C\|\Delta z\|^3$ for some $C > 0$.

The prediction error directly bounds state deviation:

$$\|\Delta z\| = \|z - z^*\| \leq \|e(t)\| \leq \sqrt{\epsilon^2 + \frac{P^2}{\alpha^2}} \tag{19}$$

When this is less than $\delta$, the quadratic term dominates since:

$$\frac{|R(\Delta z)|}{\|\Delta z\|^2} \leq C\|\Delta z\| \to 0 \tag{20}$$

The bound follows from $\lambda_{max}(H) = H_M$ and the error bound.    $\square$

These results establish quantitative bounds linking world model accuracy ($\epsilon$), control authority ($\alpha$), and perturbation magnitude ($P$) to performance. The quadratic scaling reflects the natural structure of value functions around optimal trajectories.

