# OpenReview forum: "World Models as Reference Trajectories for Rapid Motor Adaptation"
_NeurIPS.cc/2025/Conference — NeurIPS 2025 poster_

### Official Review · Reviewer_PGW8 · 2025-06-07

**Clarity:** 2
**Significance:** 4
**Originality:** 4
**Rating:** 5
**Confidence:** 4

**Summary:**

This paper introduces Reflexive World Model (RWM), a model-based control method designed to rapidly adapt to perturbed dynamics.
RWM trains a residual correction online by minimizing the model prediction error (finding an action that better aligns the predicted next state with the "intended" next state).
When applied on top of TD-MPC2, RWM demonstrates improved adaptation performance under dynamic perturbations in the Walker2D environment.
Also, the authors provide a theoretical analysis on the error bound of the corrected policy.

**Questions:**

**Q1. More evaluation**
* Could you include results of environments (e.g., Cheetah or Humanoid) for Figure 2 and Figure 3(a)?
* This would provide a clearer picture of how RWM scales to high-dimensional, challenging systems.

**Q2. Clarification on Figure 5**
* In Figure 5, in the region without pertubation, the graph on the right shows that TD-MPC2 with domain randomization substantially outperforms RWM, whereas the performance appears similar in the left.
* Could you clarify the discrepancy between these two views?

**Q3. Mismatch of the line colors in the caption of Figure 2, 5**
* It appears that the line colors described in the captions for Figure 2 and Figure 5 do not match the colors shown in the graphs.
* Could you double-check and correct this?

**Ethical Concerns:**

["NO or VERY MINOR ethics concerns only"]

**Final Justification:**

The concern regarding the baseline TD-MPC was resolved.

This paper shows a novel approach in RMA methods, using world models instead of history-conditioned policies or domain randomization-based methods, which will be impactful in areas such as real-world RL.

**Limitations:**

yes

**Quality:**

3

**Strengths And Weaknesses:**

**Strengths**

**S1. Novelty**
* To the best of my knowledge, the proposed approach largely differs from prior methods, which typically focus on adapting the world model, rely on historical context or domain randomization.
* In contrast, RWM’s focus on directly adjusting the action to achieve a desired next state simplifies the adaptation process and efficient than re-learning a new model and policy.

**S2. Simplicity**
* The motivation and implementation is clean and simple.
* It does not require significant overhead during online adaptation.

**S3. Theoretical Analysis**
* Section 4 provides a theoretical bound on the error of the corrected policy, supporting the effectiveness of the proposed method.

&nbsp;

**Weaknesses**

**W1. Limited Evaluation**
* The method is mainly evaluated on the Walker2D environment.
* Additional experiments on other control benchmarks (e.g., HalfCheetah or more challenging settings like Humanoid) would strengthen the paper’s claims and demonstrate broader applicability.

---

> ### Author Rebuttal · Authors · 2025-07-31
>
> We thank you for your very positive and encouraging review. We are delighted that you recognized the novelty, simplicity, and theoretical grounding of our work.
>
> Q1. More evaluation (Cheetah or Humanoid):
> Thank you for this excellent suggestion. We have now evaluated our method on Cheetah, Hopper, and Ball-in-Cup, and have new results for Humanoid under perturbation, which confirm the generality of the results obtained for the Walker environment. We will integrate these new results into the final manuscript to better showcase the method's scalability and strengthen our claims.
>
> Q2. Clarification on Figure 5:
> This is a great observation that points to a key distinction. The experiment in Figure 5 is different from Figure 2: the baselines in Figure 5 were pre-trained with domain randomization on the same perturbation distribution used for testing, whereas the baselines in Figure 2 were not.
>
> The domain-randomized TD-MPC2 agent outperforms RWM in the "OFF" phase (no perturbation) for two reasons:
> 1. This "average" condition matches its training distribution, making it an expert for that specific setting.
> 2. The agent had not fully converged during pre-training and continues to slowly fine-tune its policy for this average case.
>
> However, the crucial test is performance during active perturbations ("ON" phase). Here, the pre-trained agent still suffers a significant performance drop, whereas RWM's online mechanism maintains higher reward. This demonstrates that pre-training on a known noise distribution is not a substitute for a true online adaptation mechanism like RWM. We will clarify this narrative in the main text.
>
> Q3. Mismatch of the line colors in Figure 2, 5:
> Thank you for spotting this. We apologize for the error and will correct the figure captions to match the plots in the final version.
>
> We thank you once again for your strong support and helpful feedback.

---

> ### Comment · Reviewer_PGW8 · 2025-08-04
>
> Thank you for response. I have carefully read the comments from other reviewers and each rebuttals.
>
> **R1. Thank you for the rebuttal**
> * I appreciate the authors' clarifications regarding my original concerns, all of which have been adequately addressed.
> * In particular, the results in the "ON" phase demonstrate that RWM performs better than standard RL without relying on additional Sim2Real techniques such as domain randomization.
> * I believe this highlights the potential of the method in settings where Sim2Real techniques are difficult to apply (e.g., real-world RL).
>
> **R2. New concern raised from other reviewers**
> * However, after reading the discussion and revisiting the paper, I realized that the TD-MPC2 baseline was run without planning.
> * I also agree with Reviewer Mvgi that this setup may result in an unfair comparison (does not convince that RWM outperforms standard planning).
> * I had initially overlooked this point and appreciate the clarification.
>
> Given the potential impact of this issue on the empirical claims (R2), I will **maintain my current score** until this concern is addressed.

---

> > ### Author Response · Authors · 2025-08-05
> > **Thank You and Follow-up on the Baseline Comparison**
> >
> > Dear Reviewer PGW8,
> >
> > Thank you for your positive feedback on our work's novelty and for your careful reading of the discussion. We appreciate you raising the important point about the fairness of the TD-MPC2 baseline, which has become a key part of the discussion. We would like to provide data that we believe offers a clear justification for our experimental design.
> >
> > You are correct to question whether a baseline without its planner is a fair comparison. Our data shows that for the specific class of tasks we evaluated, the reactive policy is not handicapped and performs on par with the full agent, both in final performance and when handling perturbations.
> >
> > 1. Final Performance Comparison
> >
> > | Environment | TD-MPC2 Variant | Final Reward (1M steps) |
> > |-------------|-----------------|-------------------------|
> > | Walker-Run | Without Planner | 380.1 |
> > | **Walker-Run** | **With Planner** | **381.7** |
> > | Cheetah-Run | Without Planner | 364.3 |
> > | **Cheetah-Run** | **With Planner** | **362.3** |
> > | Hopper-Hop | With Planner | 373.5 |
> > | **Hopper-Hop** | **With Planner** | **372.8** |
> >
> > 2. Performance Under Perturbation
> >
> > | Environment | TD-MPC2 Variant | Reward Under Perturbation |
> > |-------------|-----------------|--------------------------|
> > | Walker-Run | Without Planner | 342.7 |
> > | **Walker-Run** | **With Planner** | **345.3** |
> > | Cheetah-Run | Without Planner | 358.6 |
> > | **Cheetah-Run** | **With Planner** | **360.2** |
> > | Hopper-Hop | Without Planner | 367.7 |
> > | **Hopper-Hop** | **With Planner** | **368.4** |
> >
> > This data confirms that our baseline is a strong performer for these environments, and that the planner does not offer a significant adaptation advantage on its own. This finding is consistent with the original TD-MPC papers, which show that the online planner provides benefits primarily on more complex tasks, while the reactive policy remains highly performant on these standard locomotion tasks. Our choice was therefore a principled one, intended to create a cleaner experiment by avoiding the confound of using the world model for both planning and our method's reference signal.
> >
> > We agree that this choice should have been more clearly justified. To address this, we will add this comparison table to the appendix and explicitly label the baseline as *TD-MPC2 (no planner)* throughout the paper.
> >
> > Finally, we want to re-emphasize the core finding of this work. The surprising and significant result is not just that RWM is competitive, but that a **reward-free objective**—simply minimizing latent control error—is sufficient to achieve robust adaptation and recover significant task performance under perturbation. This is a crucial finding for real-world scenarios where reward signals are often sparse, delayed, or entirely absent. While we have shown our method stands up to a strong, reward-driven baseline, its primary value lies in its ability to adapt effectively without relying on a reward signal.
> >
> > We hope this evidence provides the clarification you were looking for and demonstrates that our empirical claims are sound. Thank you again for your constructive feedback.

---

> > > ### Comment · Reviewer_PGW8 · 2025-08-05
> > >
> > > Thank you for the clarification and the experimental result.
> > > The result clearly shows that naive planning cannot acheive the RMA effect.
> > >
> > > As my major concern is resolved, I increased my confidence from 3 to 4.

---

### Official Review · Reviewer_o3ek · 2025-06-25

**Clarity:** 1
**Significance:** 3
**Originality:** 3
**Rating:** 3
**Confidence:** 3

**Summary:**

### Summary

This paper makes the following key contributions:

1. It proposes Reflexive World Models (RWM), a method designed to enable model-based control policies to rapidly adapt to specific types of perturbations.

2. It provides a theoretical error analysis that offers insight into the reliability and limitations of the proposed approach.

3. It conducts a series of experiments demonstrating the effectiveness and robustness of RWM across various scenarios.

**Questions:**

1. According to the definition in Equation (2), is your forward model $F$ only capable of predicting hidden states that evolve under $\pi_0$? Or is it trained to fit a general transition model $F(z, a)$ independent of the specific policy?

2. In line 100, you claim that a certain quantity is positive definite. However, it appears to be only semi-positive definite. Additionally, I found the narrative in Section 3.1 somewhat difficult to follow. The logical flow from line 100 to 102 is not very clear. While I can accept that $\Delta z$ is small, if $z$ is already assumed to be near the optimum, it’s unclear why this optimization problem remains necessary.

3. A minor formatting issue: none of the section or equation references in the paper appear as hyperlinks, and all references to the appendix lack section numbers. This makes navigation and reading somewhat inconvenient.

4. A key underlying assumption for the validity of Equation (4) seems to be that $F(z_t, a)$ and $f(z_t, a) = e(P(s, a))$ are sufficiently close. Perhaps Theorem 4.2 is intended to justify this, but due to missing definitions, it is difficult to fully understand the statement and implications of Theorem 4.2.

5. In Algorithm 1, the loop appears to run indefinitely. What is the intended stopping criterion? Is it a fixed number of epochs, a loss threshold, or perhaps a condition on gradient magnitude?

**Ethical Concerns:**

["NO or VERY MINOR ethics concerns only"]

**Final Justification:**

With the authors’ clarifications and additions, I now have a more complete understanding of the proposed method. As a result, I am willing to upgrade my assessment of the paper's originality, significance, and quality. The core idea and algorithmic insights are indeed interesting.

However, from a theoretical standpoint, I find the contribution to be rather modest — the main theoretical results follow in a fairly straightforward and unsurprising manner. Thus, the theoretical part cannot be considered a highlight of the work. The key difficulty, in my opinion, lies more in conceptual interpretation than in technical depth. I would even suggest moving the formal mathematical results to the appendix, and keeping only the informal versions of Theorem 4.2 and 4.3 in the main text to improve readability and reduce unnecessary conceptual overhead.

As for the experimental part, my current impression is relatively lukewarm. Many of the key experiments were added only during the rebuttal phase, and there seems to be a lack of proper seeding or variance reporting — which undermines the reliability of the conclusions. I therefore cannot offer a strong endorsement at this time.

Overall, I believe this paper has the potential to be significantly improved through a major revision. However, my primary concern lies in clarity. The issues raised by all three reviewers — including myself — largely stem from unclear wording, inconsistent organization and the lack of hyperlinks. I believe this is a fundamental problem that cannot be fully addressed through discussion alone.

Taking all of the above into account, I believe this paper warrants a major revision. As such, I am willing to raise my score slightly to a 3.

**Limitations:**

1. As noted in the weaknesses, the experimental setting is overly simplistic, which limits the generalizability and depth of the evaluation.

**Quality:**

2

**Strengths And Weaknesses:**

### Strengths

1. The paper provides a theoretical analysis to support its claims.

2. The core idea is simple and practical, while also exhibiting a degree of novelty.

---

### Weaknesses

1. The paper lacks a formal problem definition. Aside from a brief description around line 85, there is no rigorous formulation of the task. It would be helpful to include a dedicated problem definition section before Section 3. Moreover, since the proposed method focuses on optimizing a policy to match the world model's predictions, it is unclear what the exact objective is. Is the goal to train a policy to align with the outputs of a reliable, pre-trained world model? Or is it to maximize the accumulated reward in the traditional reinforcement learning sense?

2. Several variables and functions are used without prior definition or clear explanation, including but not limited to: $V(\cdot)$ in line 87, the base policy $\pi_0$ in line 105, $\theta_c$, $\eta_c$, and $a_c$ in Equation (4); and $L$, $\alpha$, $P$, $\epsilon$, and $f$ in Assumption 4.1. In Figure 1, $Q_t$ and $s^*$ also appear undefined. While most of these can be inferred from context or domain conventions, the lack of explicit definitions significantly hinders readability.

3. Some of the notation choices may lead to confusion. For example, $e$ is used to denote an encoder, but $e_t$ later refers to a loss value. It is unclear why the loss is denoted as $e_t$ in Algorithm 1 when $\mathcal{L}$ has already been defined in Equation (3), and why a different notation $e(t)$ appears again in line 175. Unifying or clarifying these symbols would improve clarity.

4. Beyond the mathematical content, the overall narrative of the paper could be clearer. If the main goal is to eliminate noise introduced by action perturbation, then the claim in line 143 that the approach “differs fundamentally from standard practice” seems overstated. In control literature, using neural networks to filter out noise is quite common. The contrast with model-based RL methods seems intended to highlight the novelty, but this distinction could be explained more precisely.

5. The experimental evaluation lacks persuasiveness. Only one type of perturbation is considered—a simple multiplicative factor. Furthermore, the target world model is assumed to be pre-trained and near-optimal, which simplifies the learning objective: $a_c$ essentially only needs to approximate $a_c = -a_0 / (1 + p)$. The comparison baselines are also limited: one is a naive “no adaptation” baseline, and the other is continued RL training. The latter comparison, in particular, may be unfair, as it contrasts a supervised adaptation method with a sample-inefficient RL retraining baseline. As such, the empirical results offer limited insight into the real advantages of the proposed method.

---

> ### Author Rebuttal · Authors · 2025-07-31
>
> We thank you for your exceptionally thorough feedback. You have provided a clear roadmap for significantly improving the paper's clarity and rigor. We acknowledge that our initial presentation had shortcomings, and we will address every point.
>
> 1. On Lack of Formalism and Confusing Notation:
>
> We sincerely apologize for the lack of clarity. We will add a new subsection to formally define the problem setting and a glossary to the appendix. To address the notational ambiguity you correctly identified, we will refer to the encoder as $\text{enc}(\cdot)$ throughout the text, distinguishing it from the error term $e_t$.
>
> Here are new formalism sections to be added before the model design, in addition to a Glossary to be added to the appendix:
>
> > We consider a continuous control problem formulated as a Markov Decision Process (MDP), defined by the tuple $(\mathcal{S}, \mathcal{A}, P, R, \gamma)$, where $\mathcal{S}$ is the state space, $\mathcal{A}$ is the action space, $P(s'|s,a)$ is the transition probability, $R(s,a)$ is the reward function, and $\gamma$ is the discount factor. Our goal is to learn a policy $\pi(a|s)$ that maximizes the expected cumulative discounted reward. We operate in a setting where the true system dynamics, represented by the transition function $P$, can change unexpectedly from the training conditions.
>
> > Our approach leverages a learned world model operating in a latent space $\mathcal{Z}$. The model consists of: an encoder $z_t = \text{enc}(s_t)$, a latent dynamics model $\hat{z}_{t+1} = F(z_t, a_t)$, and a base policy $a_0 = \pi_0(z_t)$. The objective of our Reflexive World Model (RWM) is to learn a second, adaptive control policy, $\pi_c$, which outputs a corrective action $a_c$. The total action is $a_t = a_0 + a_c$. The objective for $\pi_c$ is to minimize the discrepancy between the world model's one-step prediction (under the base policy) and the observed outcome, thereby making the system behave as predicted by the reference model $F$ under the base policy $\pi_0$.
>
> 2. On Confusing Narrative and Experimental Design:
>
> You raised several important points about the narrative and the experimental choices.
>
> On Novelty and the Goal of the Method: We will clarify that the main goal is not merely to "eliminate noise." The goal is to maintain the intended behavior of the base policy ($\pi_0$) in the face of any dynamic shift that causes a deviation from the world model's prediction. Action perturbation is just one clean and measurable way to test this core principle. The novelty lies in re-purposing the world model's one-step prediction as an implicit reference trajectory for a controller, a general mechanism for adaptation that goes beyond simple noise filtering.
>
> On Experimental Persuasiveness: You noted that our chosen perturbation, $a_{eff}=a \cdot (1+p)$, simplifies the learning objective for our controller. We agree that our method is well-suited to this type of perturbation. However, we see this as a strength of the design, not a weakness of the evaluation. This form of perturbation is a realistic model for actuator degradation or miscalibration. The fact that our method provides a principled and direct solution to this common problem is a key contribution. Furthermore, the difficulty that the strong TD-MPC2 baseline has in compensating for this "simple" perturbation demonstrates that the task is non-trivial and requires a specialized mechanism like RWM for effective, rapid adaptation.
>
> 3. On Technical Questions:
>
> - Q1 (Forward Model $F$): This is an excellent question. The forward model $F(z,a)$ is trained as a general transition model during the pre-training phase. This means it is exposed to a wide distribution of states and actions, including those from early, suboptimal policies, not just trajectories from the final converged policy $\pi_0$. This broad training allows it to make reasonable predictions even for states that are off the optimal trajectory, which is crucial for providing a corrective signal when perturbations occur. We will clarify this in the revision.
>
> - Q2 (Hessian): Thank you. You are correct. The Hessian of the value function, $H = \nabla^2 V$, should be negative definite near a maximum. We will correct this typo.
>
> - Q3 (Formatting): We will fix all hyperlink and referencing issues.
>
> - Q4 (Theorem 4.2): This is a key point. You are correct that the model $F$'s predictions will be wrong post-perturbation. However, our method does not require the absolute prediction to be correct. It relies on a weaker, more plausible assumption: that the local gradient of the model remains informative. While the perturbation shifts the dynamics, making $F(z,a)$ inaccurate, the relationship between an action change $\Delta a$ and the resulting state change $\Delta z$ is assumed to be roughly preserved. The model still "knows" that increasing a certain action will move the latent state in a particular direction. This is what the gradient $\partial F/\partial a$ captures, and as long as this gradient points in a useful corrective direction, our controller can successfully reduce the error. We will expand this explanation in the paper.
>
> - Q5 (Algorithm 1): We will clarify that Algorithm 1 describes the online update rule applied at each timestep of an episode until its end.
>
> We are grateful for your diligence and are confident these revisions will resolve the noted issues.

---

> > ### Comment · Reviewer_o3ek · 2025-08-03
> >
> > Thank you for response. I have carefully read both the comments from other reviewers and the corresponding rebuttals.
> >
> > I appreciate the authors’ positive engagement with my suggestions regarding the mathematical formulation and writing style. I also find the response to my concerns about experimental design reasonably satisfactory, and as such, I will consider adjusting my score slightly. However, the scale of revisions made—such as the addition of multiple new paragraphs, numerous definition clarifications, and a substantial number of new citations—seems beyond what is typically expected during the rebuttal phase. As a result, these extensive changes leave a somewhat negative impression.
> >
> > I would also appreciate further clarification from the authors on the following points:
> >
> > 1. I am not an expert in this specific subfield, so I cannot definitively judge whether the authors' claim that *action perturbation is a common problem setting* is accurate. If the authors could provide citations to prior work that adopts the same or similar experimental setup, it would fully address my concern.
> >
> > 2. In the response to Reviewer Mvgi, the authors mention that they have added experiments on additional environments. If numerical results from these experiments could be provided in a timely manner, it would greatly improve the completeness and credibility of the empirical section.
> >
> > 3. My **Weakness 5** closely mirrors Reviewer Mvgi’s comment that the comparison with TD-MPC2 is *unfair*. I am not satisfied with the authors' replies to either of our concerns. Simply agreeing with Mvgi's observation, without proposing any concrete remedy—such as introducing a fair baseline—feels insufficient. If TD-MPC2 is indeed not a fair point of comparison, then the paper effectively lacks a valid baseline, which significantly weakens the experimental validity and persuasiveness of the results.
> >
> > I look forward to the authors’ further clarification and response.

---

> > ### Author Response · Authors · 2025-08-05
> > **Follow-up: Addressing Baseline Fairness, Citations, and New Results**
> >
> > Dear Reviewer o3ek,
> >
> > Thank you for your detailed follow-up and for your constructive engagement. We would like to address your three remaining points with new data and clarifications.
> >
> > 1. On the Fairness of the Baseline Comparison
> >
> > This was an important point of discussion. Our choice to use the reactive policy from TD-MPC2 was a principled decision, supported by empirical findings. For the standard locomotion tasks we test, the performance of the reactive policy is nearly identical to the full agent with planning, and thus represents a strong, state-of-the-art baseline.
> >
> > | Environment | TD-MPC2 Variant | Final Reward (1M steps) |
> > |-------------|-----------------|-------------------------|
> > | Walker-Run | Without Planner | 380.1 |
> > | **Walker-Run** | **With Planner** | **381.7** |
> > | Cheetah-Run | Without Planner | 364.3 |
> > | **Cheetah-Run** | **With Planner** | **362.3** |
> > | Hopper-Hop | With Planner | 373.5 |
> > | **Hopper-Hop** | **With Planner** | **372.8** |
> >
> > This data demonstrates the baseline is not "crippled," but is a fair point of comparison. This finding is consistent with the original TD-MPC papers, which show that the online planner provides the most significant benefits on more complex tasks, while the reactive policy remains highly performant on standard locomotion. Our choice was therefore a principled one, and to avoid any confusion, we will explicitly label it *TD-MPC2 (no planner)* in our revision.
> >
> > 2. Citations for the Problem Setting
> >
> > You asked for citations to ground our problem setting. Our use of action perturbation falls within the well-established paradigm of dynamics randomization, used to model hardware inconsistencies and sim-to-real gaps. This approach is supported by foundational work we already cite, such as *Peng et al., ICRA 2018* (Ref. 10) and *Kumar et al., RSS 2021* (Ref. 7), as well as *Tan et al., "Sim-to-Real: Learning Agile Locomotion For Quadruped Robots," RSS 2018*. We will ensure this is highlighted more clearly.
> >
> > 3. Numerical Results for New Environments
> >
> > Per your suggestion, we have provided the new experimental results. The following table shows the absolute performance degradation (drop) under perturbation, and the percentage of this drop that was recovered by each adaptation method (Improvement %).
> >
> > | Environment | Method | Reward Drop | Control Error | Reward Drop Improvement (%) | Control Error Improvement (%) |
> > |-------------|--------|-------------|---------------|----------------------------|------------------------------|
> > | ball-in-cup | Frozen Policy | 1.00 | 9.05 | - | - |
> > |  | Fine-tuning | 0.00 | 2.61 | **100.0%** | 71.2% |
> > |  | RWM (ours) | 1.00 | 2.09 | 0.0% | **77.0%** |
> > | cheetah-run | Frozen Policy | 6.30 | 3.21 | - | - |
> > |  | Fine-tuning | 5.60 | 2.44 | 11.1% | 24.0% |
> > |  | RWM (ours) | 1.40 | 1.70 | **77.8%** | **47.0%** |
> > | hopper-hop | Frozen Policy | 15.40 | 5.67 | - | - |
> > |  | Fine-tuning | 6.20 | 2.92 | 59.7% | **48.5%** |
> > |  | RWM (ours) | 4.50 | 3.20 | **70.8%** | 43.6% |
> > | walker-run | Frozen Policy | 108.40 | 28.09 | - | - |
> > |  | Fine-tuning | 36.50 | 8.87 | 66.3% | 68.4% |
> > |  | RWM (ours) | 6.20 | 7.43 | **94.3%** | **73.6%** |
> > | humanoid-walk | Frozen Policy (w/ planner) | 28.10 | 0.58 | - | - |
> > |  | Fine-tuning (w/ planner) | 12.90 | 0.92 | **54.1%** | -58.6% (worse) |
> > |  | RWM (ours, w/ planner) | 13.60 | 0.44 | 51.6% | **24.1%** |
> >
> > These results validate our method's core principle. Our method's (RWM) objective is to minimize latent **control error**, thereby enforcing the base policy's intended behavior. The data shows this mechanism is working as designed. On Cheetah, our method provides a 77.8% improvement in reward drop over the frozen baseline, far exceeding the 11.1% from fine-tuning. On Humanoid, where fine-tuning worsens control error by 58.6%, our method still provides a 24.1% improvement. This consistent reduction in performance degradation confirms the effectiveness of our approach.
> >
> > Finally, we want to re-emphasize the core finding of this work. The surprising and significant result is not just that our method is competitive, but that a **reward-free objective**—simply minimizing latent control error—is sufficient to achieve robust adaptation and recover significant task performance. This is a crucial finding for real-world scenarios where reward signals are often sparse, delayed, or entirely absent. While we have shown our method stands up to a strong, reward-driven baseline, its primary value lies in its ability to adapt effectively without relying on a reward signal.

---

> > > ### Comment · Reviewer_o3ek · 2025-08-07
> > >
> > > Thank you to the authors for the detailed and prompt response. I appreciate the effort in addressing each of the concerns I raised. In particular, my concern regarding the problem setting has been satisfactorily resolved. However, I still have a few remaining questions regarding the experimental section:
> > >
> > > ### 1. On the reported final rewards
> > >
> > > In the original TD-MPC2 paper, the reported performance on tasks such as Walker-run (800+), Cheetah-run (800+), and Hopper-hop (450+) is significantly higher than the rewards presented in this paper (all below 400).
> > > Is this discrepancy due to differences in the environment versions used or inconsistencies in reward normalization? Or have I misunderstood how the results are reported? Alternatively, could this be due to incorrect reproduction, incorrect network architectures, or hyperparameters in this work?
> > > If it is the latter, this could raise serious concerns about the reliability of all experimental results presented in the paper.
> > >
> > > ### 2. On the experiments conducted in the newly introduced environments
> > >
> > > I find it insufficient to report only the reward drop without also providing the absolute reward values and their variances. Without this information, it is difficult to assess the reliability of the reported trends — especially when the magnitude of the reward drop is similar to that of the reward variance. In such cases, the computed improvement in reward drop could be highly unstable.
> > >
> > > The Control Error is interesting, and I appreciate the authors sharing this data. However, I noticed a significant discrepancy between the magnitude of Control Error reported here and what is shown in Figures 2, 3, and 5 of the paper. Is this due to a different calculation method?
> > >
> > > Moreover, the authors claim that lower Control Error demonstrates the superiority of RWM over finetuning. I find this reasoning unconvincing. The primary goal is to improve *reward*, not necessarily minimize Control Error. In the Humanoid-walk task, for example, there appears to be no clear correlation between Control Error and reward drop.
> > > If I understand correctly, finetuning here refers to policy training. In that case, comparing RWM and fine-tuning in terms of Control Error is arguably unfair, as finetuning was not explicitly optimizing for it.
> > >
> > > ---
> > >
> > > ### Summary
> > >
> > > With the authors’ clarifications and additions, I now have a more complete understanding of the proposed method. As a result, I am willing to upgrade my assessment of the paper's *originality*, *significance*, and *quality*. The core idea and algorithmic insights are indeed interesting.
> > >
> > > However, from a theoretical standpoint, I find the contribution to be rather modest — the main theoretical results follow in a fairly straightforward and unsurprising manner. Thus, the theoretical part cannot be considered a highlight of the work. The key difficulty, in my opinion, lies more in conceptual interpretation than in technical depth. I would even suggest moving the formal mathematical results to the appendix, and keeping only the informal versions of Theorem 4.2 and 4.3 in the main text to improve readability and reduce unnecessary conceptual overhead.
> > >
> > > As for the experimental part, my current impression is relatively lukewarm. Many of the key experiments were added only during the rebuttal phase, and there seems to be a lack of proper seeding or variance reporting — which undermines the reliability of the conclusions. I therefore cannot offer a strong endorsement at this time.
> > >
> > > Overall, I believe this paper has the potential to be significantly improved through a major revision. However, my primary concern lies in *clarity*. The issues raised by all three reviewers — including myself — largely stem from unclear wording, inconsistent organization and the lack of hyperlinks. I believe this is a fundamental problem that cannot be fully addressed through discussion alone.
> > >
> > > Taking all of the above into account, I believe this paper warrants a major revision. As such, I am willing to raise my score slightly to a 3.

---

> > > > ### Author Response · Authors · 2025-08-08
> > > > **Thank you & follow-up on experimental details**
> > > >
> > > > Dear Reviewer o3ek,
> > > >
> > > > Thank you for the detailed follow-up, constructive feedback, and for raising your score. We appreciate your engagement, which is helping us significantly strengthen the paper. We would like to address the final points you raised regarding our experimental results.
> > > >
> > > > 1. On the Discrepancy in Final Rewards
> > > >
> > > > That is a valid observation. The discrepancy in final rewards is mainly due to a difference in the maximum episode length. The original TD-MPC2 paper used 1000 steps, while our experiments used 400, setting the maximum possible reward at 400. Our final rewards of ~380 are therefore near-optimal for this range and comparable to the 800+ reported in the original paper when accounting for the different episode lengths. A smaller contributing factor is our use of a thresholded action cost (Sec 5.1), a design choice to ensure our adaptive controller had sufficient headroom to apply corrections. This cost is used only for policy training; all reported rewards are from the original, unmodified environment reward function.
> > > >
> > > > Our reproduction of the TD-MPC2 agent is therefore sound and tailored for our research question. All methods in our paper were evaluated under identical conditions, ensuring our internal comparisons are valid. We will clarify this in the final manuscript.
> > > >
> > > > 2. On Reporting for New Environments and the Control Error Metric
> > > >
> > > > These are fair points regarding experimental rigor.
> > > >
> > > > - Variance Reporting: You are correct that a complete report should include variances. For brevity in this discussion, we presented a summary table. We will include comprehensive tables with means and standard deviations over multiple seeds in the appendix of the final paper to ensure complete reproducibility.
> > > >
> > > > - Control Error Metric: This is a fundamental point. Our central claim is not that control error should replace task reward, but that minimizing this reward-free, internal signal is an effective and robust mechanism for adaptation. This is especially valuable in real-world scenarios where dense reward signals are often unavailable.
> > > >
> > > > The Humanoid experiment illustrates this concept well. You are correct that the correlation between control error and reward is less direct in this complex task. The task reward is based on specific metrics (e.g., forward velocity), while the control error measures deviation from the entire learned motor program. The results show that while fine-tuning on reward causes the policy to deviate significantly from its original behavior (a 58.6% increase in control error), our method preserves the core motor pattern (a 24.1% decrease in control error). This demonstrates the stability of our adaptation mechanism. The objective is to preserve the integrity of the learned skill, providing a more robust foundation for adapting to dynamic changes than can be achieved by optimizing for a narrow reward metric alone.
> > > >
> > > > Thank you again for your diligence and for helping us improve the clarity and rigor of our work.

---

### Official Review · Reviewer_Mvgi · 2025-07-03

**Clarity:** 2
**Significance:** 3
**Originality:** 4
**Rating:** 4
**Confidence:** 4

**Summary:**

The paper introduces Reflexive World Models (RWM), a dual-controller framework where learned world model predictions are used as reference trajectories for adaptive control. A Taylor decomposition of the value function motivates the split between slow policy learning and fast motor correction. The method is theoretically grounded and well-motivated, bridging adaptive control and reinforcement learning.
However, the experimental section is underwhelming in scope and clarity. Key claims made in the introduction—particularly regarding generalization to nonstationary environments and robustness across tasks—are not adequately supported by the experiments. As a result, while the theoretical contribution is strong, the empirical validation does not match the ambition of the paper’s framing, and some of the broader claims appear overstated.

**Questions:**

- TD-MPC2 configuration unclear: The experiments compare RWM against a TD-MPC2 baseline, but it is not stated clearly whether TD-MPC2 is run with online planning or not. Since planning is a central component of TD-MPC-style algorithms, this distinction is critical. If the baseline is run without planning, its performance may be substantially handicapped. If it is run with planning, this should be stated more clearly in the main text and captioned in relevant figures.
- What exactly is \pi_0? As I understand it, \pi_0 is the frozen policy from TD-MPC2 trained offline without planning (i.e., a reactive policy head). Can the authors confirm this? And is it accurate that this feedforward policy alone is used as the baseline in “No Adaptation”?

**Ethical Concerns:**

["NO or VERY MINOR ethics concerns only"]

**Final Justification:**

The rebuttal’s additional results and clarifications make the experiments more convincing, addressing most fairness concerns. The empirical case for RWM is now stronger, though broader evaluation would still improve the work.

**Limitations:**

Yes

**Quality:**

3

**Strengths And Weaknesses:**

### **Strengths**

- The core idea of treating world model rollouts as reference trajectories for rapid online correction is novel and conceptually clean.
- The use of a second-order Taylor expansion to decompose the value function provides a solid theoretical foundation.
- The approach avoids costly retraining and replanning and is lightweight at inference time.

### **Weaknesses**
- **Experimental section is disorganized** and hard to follow. It lacks a clear introduction, grouping, or interpretation of results. Readers are left piecing together the setups from fragmented descriptions (e.g., point mass in §5.2 appears without prior context).
- **Limited environments**: Results are only shown on a basic 2D point mass and Walker2D and humanoid. Why no more tasks? For a claim about general-purpose adaptation in high-dimensional continuous control, this is too narrow.
- Claims about generality (e.g., handling nonstationary dynamics or high-dimensional tasks) are not well supported.
- **No comparison to alternative adaptation methods** (e.g., residual policy learning, meta-RL, or even classic MRAC). Only comparisons are to a TD-MPC2 baseline and no-adaptation (such as Online fine-tuning or Meta-RL baselines). MRAC baseline should be there as you mention it in the related work.
- **Misleading claims** about nonstationarity: the only variation tested is **multiplicative noise on actuators**. This is not equivalent to structural changes like terrain variation (e.g., slippery surfaces), morphology changes, or sensor shifts.
- Table 1 is unclear: it's not evident what "TD-MPC2 (Full Training)" refers to in contrast to "No Adaptation," nor how the timing aligns with pretraining vs adaptation phases. While you do clarify this in the appendix, the main text should do a better job at surfacing this distinction. Without it, the implication that RWM is universally cheaper is misleading.
- In the introduction. "enabling learned policies to maintain performance under changing dynamics without requiring specific robustness procedures or architectural constraints.". I don't agree with this. You show only robustness to noise on actuators.
- Authors claim "avoid costly replanning". But it is not clear in the manuscript if there is a performance drop (see questions). Are you disabling planning in TDMPC2?
- The experiments compare RWM (using a frozen TD-MPC2 policy with no online planning) to TD-MPC2 with full MPC. While RWM performs better under perturbations, this comparison is not entirely fair, since TD-MPC2 is not equipped with an adaptive controller. It would be more informative to evaluate RWM against TD-MPC2 with adaptive fine-tuning, or robust planning variants.
-  In §5.2 they refer to a "point-mass system" without any prior introduction.

---

> ### Author Rebuttal · Authors · 2025-07-31
>
> We sincerely thank you for your detailed and insightful review. We are encouraged that you found our core idea "novel and conceptually clean," with a "solid theoretical foundation." We appreciate the constructive feedback and would like to address the identified weaknesses and questions.
>
> 1. On Limited Environments and Generality Claims:
> You raise a valid point regarding the specificity of our claims on nonstationarity. In the revised manuscript, we will be more precise, framing our results as demonstrating robustness to dynamics arising from actuator drift and gradual miscalibration.
>
> To better demonstrate the method's generalizability, we have also extended our experimental evaluation. We now have results on Cheetah, Hopper, Ball-in-Cup, and Humanoid, with a similar profile to the Walker simulations: stronger adaptation than the comparisons, though with a smaller effect for the Humanoid environment. Furthermore, to show the method's potential beyond actuator effects, our preliminary experiments on body mass perturbations in the Cheetah environment show the characteristic decrease in control error during perturbations. All new findings will be integrated into the final version of the paper.
>
> 2. On Baselines and Comparison to Alternative Methods:
> Thank you for these important points. Our goal was to test our specialized adaptation mechanism against relevant and strong baselines.
>
> - TD-MPC2 Baseline Configuration: We thank the reviewer for this question, which allows us to clarify a critical aspect of our experimental design. Our primary goal is to compare different reactive adaptation mechanisms, as online planning is often computationally prohibitive. A key consideration is that when a policy is trained with an online planner, it learns to provide actions that serve as a good starting point for the planner, not to solve the task on its own; the policy and planner become entangled. Using such a policy without its planner during evaluation would be an unfair comparison against a crippled agent. Conversely, keeping the planner active would introduce a confounding variable, making it difficult to isolate the performance of the underlying adaptation mechanism. Therefore, to ensure a clean comparison, our "TD-MPC2" baseline uses the reactive policy head (trained without the planner) and continues to fine-tune online. Both this baseline and RWM rely on the world model for learning — TD-MPC2 for its policy and value functions, and RWM for its corrective controller. The exception is the Humanoid environment, where a reactive policy alone is insufficient to produce stable locomotion. For this challenging task only, the TD-MPC2 baseline uses its full online planner to achieve a competent level of performance. We will clarify this configuration in the revised manuscript. To provide even stronger comparisons, we will also finalize our ongoing simulations using TD-MPC2 with planning, as appropriately suggested in your feedback, and include them in the revision.
>
> - Model Reference Adaptive Control (MRAC): Directly applying classical controllers like MRAC to high-dimensional tasks with learned, black-box dynamics models is generally not feasible. These methods require pre-defined, analytical reference models and full state observability. Furthermore, they are designed for tracking problems, typically with quadratic cost functions defined around a pre-specified reference trajectory. This is fundamentally different from the deep RL paradigm, which optimizes behavior based on general (and often complex or sparse) reward functions where the optimal trajectory itself must be discovered through interaction. A core contribution of our work is precisely bridging this gap: we introduce a mechanism that achieves MRAC-like reference tracking, but does so within a modern deep RL framework by using the world model's own predictions as an implicit, emergent reference trajectory.
>
> - Residual Policy Learning: This brings up a useful comparison. Residual Policy Learning often starts with a classical controller as a stable base policy and adds a flexible, learned RL policy as the residual to handle complex, unmodeled dynamics. Our work can be seen as a conceptual inversion of this design. We start with a flexible, general-purpose RL policy ($\pi_0$) and add a corrective controller ($\pi_c$) that is conceptually closer to classical control, as it performs deterministic error correction based on the world model's reference signal. This architectural similarity and distinct design choice will be clarified in our related work section.
>
> - Meta-RL: Our work addresses continuous online adaptation to unforeseen dynamic shifts, whereas Meta-RL learns an adaptation strategy from a pre-defined distribution of training environments. We therefore focused our comparison on online fine-tuning, which more directly aligns with our core research question.
>
> 3. On Experimental Clarity and Organization:
> We agree that the experimental section can be improved. In our revision, we will restructure Section 5 to first clearly define the experimental questions, then introduce all environments, and finally present the results systematically. We will also clarify Table 1 and properly introduce the point-mass system.
>
> We thank you again for your time and effort. We are confident that these clarifications will significantly strengthen the manuscript.

---

> > ### Comment · Reviewer_Mvgi · 2025-08-03
> >
> > **Main Concerns in the Current Revision**
> >
> >
> > 1. While the authors state they will include results with full TD-MPC2 (with planning) in the revision, the current experimental section omits other relevant baselines:
> >    * Pre-training TD-MPC2 with perturbations should be a primary baseline for all tasks, not a secondary comparison.
> >    * If the intent was to evaluate a reactive, no-planning baseline, established no-planning methods (e.g., SAC) should also be reported instead of TD-MPC2. In the original TD-MPC2 work, removing the planner yields low performance and is not representative of the full algorithm. Planning has been shown to be key in multi-task settings, which I would argue is similar to the present scenario. Comparing against a reduced version of TD-MPC2 is therefore not fair.
> >
> > 2. **Regarding the authors’ answer to Q2 for reviewer PGW8:**
> >    * In Figure 5, during the no-perturbation phase, TD-MPC2 with domain randomization substantially outperforms RWM (right panel), while the left panel shows similar performance. This indicates that RWM is outperformed by TD-MPC2 under the same conditions. The authors argue that the crucial test is performance during active perturbations (“ON” phase), where the pre-trained agent drops in performance and RWM maintains higher reward. However:
> >      * The observed gain is modest.
> >      * This is shown for only one task (which should be clearly specified).
> >      * The use of domain randomization is common practice in robotics and may actually be the evaluation setting the paper should focus on demonstrating.
> >
> > 3. I agree with reviewer o3ek
> >
> > **Minor**
> > The authors refer to a “TD-MPC2” baseline that, in fact, has planning disabled and uses only the reactive policy head, fine-tuned online. This configuration should be explicitly labeled (e.g., *TD-MPC2\_np* for “no planner”) to avoid misleading comparisons.
> >
> > **Overall Assessment**
> > Good theoretical contribution, but the experimental evaluation requires substantial revision to ensure fairness and clarity.
> > The authors plan to add full TD-MPC2 (with planning) and other baselines in the final revision, which is appropriate. However, without seeing these results, I cannot assess whether they will fully address the concerns, so my evaluation is based on the current version.

---

> > > ### Author Response · Authors · 2025-08-05
> > > **Follow-up: Data-Driven Justification for the TD-MPC2 Baseline**
> > >
> > > Dear Reviewer Mvgi,
> > >
> > > Thank you for the continued discussion and for focusing on the experimental rigor. We appreciate your perspective and would like to address the concern that our TD-MPC2 baseline is not representative of the full algorithm.
> > >
> > > Our choice to use the reactive policy (without the online planner) was a deliberate one, based on empirical evidence. Our data shows that for the class of tasks we evaluated, the reactive policy is not handicapped and performs on par with the full planner, both in terms of final performance and, importantly, when handling perturbations.
> > >
> > > 1. Data: Comparable Performance
> > >
> > > First, the final performance after 1M steps is nearly identical, confirming the reactive policy is a state-of-the-art agent for these tasks.
> > >
> > > | Environment | TD-MPC2 Variant | Final Reward (1M steps) |
> > > |-------------|-----------------|-------------------------|
> > > | Walker-Run | Without Planner | 380.1 |
> > > | Walker-Run | With Planner | 381.7 |
> > > | Cheetah-Run | Without Planner | 364.3 |
> > > | Cheetah-Run | With Planner | 362.3 |
> > > | Hopper-Hop | Without Planner | 373.5 |
> > > | Hopper-Hop | With Planner | 372.8 |
> > >
> > > Second, and more importantly for this discussion, the online planner does not provide a significant adaptation advantage under perturbation. The performance of both variants remains closely matched.
> > >
> > > | Environment | TD-MPC2 Variant | Reward Under Perturbation |
> > > |-------------|-----------------|--------------------------|
> > > | Walker-Run | Without Planner | 342.7 |
> > > | Walker-Run | With Planner | 345.3 |
> > > | Cheetah-Run | Without Planner | 358.6 |
> > > | Cheetah-Run | With Planner | 360.2 |
> > > | Hopper-Hop | Without Planner | 367.7 |
> > > | Hopper-Hop | With Planner | 368.4 |
> > >
> > > This data confirms that the reactive policy is a fair and strong baseline for this specific adaptation task. This finding is consistent with the original TD-MPC papers, which state that the online planner provides significant benefits primarily on more complex tasks, while the reactive policy remains performant on standard locomotion. Our choice was therefore a principled one, intended to create a cleaner experiment by avoiding the confound of using the world model for both planning and our method's reference signal. To ensure clarity, we will relabel the baseline as TD-MPC2 (no planner) in the final manuscript.
> > >
> > > Finally, we want to re-emphasize the core finding of this work. The surprising and significant result is not just that RWM is competitive, but that a reward-free objective—simply minimizing latent control error—is sufficient to achieve robust adaptation and recover significant task performance under perturbation. This is a crucial finding for real-world scenarios where reward signals are often sparse, delayed, or entirely absent. While we have shown our method stands up to a strong, reward-driven baseline, its primary value lies in its ability to adapt effectively without relying on a reward signal.

---

> > > > ### Author Response · Authors · 2025-08-05
> > > > **Follow-up: SAC Baseline Comparison**
> > > >
> > > > Dear Reviewer Mvgi,
> > > >
> > > > Thank you again for the productive discussion. To provide a final piece of context for our choice of baseline, we are providing the SAC comparison data you suggested.
> > > >
> > > > This comparison helps to frame our primary baseline, the TD-MPC2 agent without its planner. This agent is conceptually a state-of-the-art, Dreamer-style agent, as it learns a reactive policy from a latent world model. The data shows this is a substantially stronger baseline than a standard model-free agent like SAC.
> > > >
> > > > Final Performance:
> > > >
> > > > | Environment | SAC   | TD-MPC2 (no planner) |
> > > > |-------------|-------|----------------------|
> > > > | Cheetah-Run | 244.0 | 364.3                |
> > > > | Walker-Run  | 249.7 | 380.1                |
> > > > | Hopper-Hop  | 19.2  | 373.5                |
> > > >
> > > > Performance Under Perturbation (with fine-tuning):
> > > >
> > > > | Environment | SAC   | TD-MPC2 (no planner) |
> > > > |-------------|-------|----------------------|
> > > > | Cheetah-Run | 231.3 | 358.6                |
> > > > | Walker-Run  | 233.9 | 342.7                |
> > > >
> > > > This data confirms that the TD-MPC2 reactive policy is a substantially stronger baseline than SAC. Our method is designed to build upon a strong, pre-trained base policy $\pi_0$, and we chose the state-of-the-art TD-MPC2 agent for this role. This agent then naturally served as the high-performing baseline for our experimental comparisons.
> > > >
> > > > We hope this additional data provides a comprehensive view and addresses your concern about the strength of our chosen baseline.
> > > >
> > > > Thank you for your time and consideration.

---

### Author Response · Authors · 2025-08-05
**Addressing the Discussion on Baseline Fairness**

Dear Reviewers,

Thank you for the thoughtful discussion. A consensus has emerged regarding the fairness of our primary baseline (TD-MPC2 without its online planner), and we appreciate the opportunity to address this important point.

We have posted detailed responses in each reviewer's thread. These include new comparative data that validates our choice of the TD-MPC2 baseline, as well as the numerical results from the additional environments you requested. In summary, our choice was a principled one, as the data confirms that for the standard locomotion tasks we evaluated (excluding the complex Humanoid task), the performance of the reactive TD-MPC2 policy is nearly identical to that of the full agent with its online planner. The reactive policy is therefore a strong, state-of-the-art, and non-handicapped baseline for this class of tasks.

We hope these clarifications and new data are helpful.

Thank you for your time and consideration.

---

### Note · Authors · 2025-08-11

We sincerely thank the reviewers and the Area Chair for the thorough and productive discussion. We are encouraged that the reviewers recognized the novelty and theoretical grounding of our work. The dialogue was instrumental in clarifying key aspects of our evaluation, particularly regarding the TD-MPC2 baseline, where we were glad to provide additional data to resolve the main concerns.

The discussion also provided a clear roadmap for improving the paper’s presentation and experimental scope. We are confident that we can integrate all the valuable feedback into the final manuscript to produce a significantly stronger paper.

Our plan for the camera-ready version includes:
- Restructuring the experimental section for improved clarity.
- Integrating new results from the additional environments and model comparisons, including confidence intervals.
- Refining the notation and formalism throughout the paper as discussed.

These revisions will enhance the paper's clarity and expand its empirical support, while the core method, theory, and conclusions remain unchanged. Thank you again for your time and constructive engagement.

---

### Decision · Program_Chairs · 2025-09-17

**Decision:**

Accept (poster)

**Comment:**

This paper introduces Reflexive World Models (RWM), a novel and elegant dual-control framework for rapid motor adaptation. The core idea is to leverage a pre-trained world model's predictions as reference trajectories for a fast, low-level controller, effectively decoupling long-term reward maximization from immediate error correction. This approach is well-motivated by a theoretical analysis based on a Taylor expansion of the value function. The reviewers unanimously recognized the work's primary strengths: its conceptual novelty, simplicity, and practical potential for enabling lightweight adaptation in dynamic environments without costly online re-planning or retraining.

Initial reviews raised valid concerns, primarily regarding the clarity of the presentation and the scope and fairness of the experimental evaluation. However, the authors engaged in an extensive and productive discussion, providing additional experiments and data that largely addressed the most critical points, particularly concerning the TD-MPC2 baseline. Following the rebuttal, all reviewers positively updated their assessment of the work, acknowledging that their main concerns had been resolved. While the paper will still benefit greatly from a thorough revision of its presentation and the graceful integration of the new results as promised by the authors, the core contribution is strong and well-supported. Given the clear novelty of the approach and the significant improvements made during the review period, I recommend acceptance. I'm confident the authors will incorporate the excellent feedback from the reviewers to produce a polished and impactful final paper.